Relationship between ROS production, MnSOD activation and periods of fasting and re-feeding in freshwater shrimp Neocaridina davidi (Crustacea, Malacostraca)

Włodarczyk Agnieszka 1
Wilczek Grażyna 2
Wilczek Piotr 3
Student Sebastian 4
Ostróżka Anna 1
Tarnawska Monika 2
Rost-Roszkowska Magdalena magdalena.rost-roszkowska@us.edu.pl 1
1 Department of Animal Histology and Embryology, University of Silesia in Katowice , Katowice , Poland
2 Department of Animal Physiology and Ecotoxicology, University of Silesia in Katowice , Katowice , Poland
3 Bioengineering Laboratory, Heart Prosthesis Institute , Zabrze , Poland
4 Faculty of Automatic Control, Silesian University of Technology , Gliwice , Poland
Hauser-Davis Rachel Ann
Electronic publication date: 2019 Sep 11
Publication date: 2019
Volume: 7
Electronic Location ID: e7399
Received 2019 Mar 26; Accepted 2019 Jul 2
Copyright: ©2019 Włodarczyk et al.
Copyright year: 2019
Copyright holder: Włodarczyk et al.
License: This is an open access article distributed under the terms of the Creative Commons Attribution License, which permits unrestricted use, distribution, reproduction and adaptation in any medium and for any purpose provided that it is properly attributed. For attribution, the original author(s), title, publication source (PeerJ) and either DOI or URL of the article must be cited.
License URL: https://creativecommons.org/licenses/by/4.0/

Keywords: Digestive system, Midgut, Starvation, Free radicals, Epithelium

Funding: The authors received no funding for this work.

==============================
The middle region of the digestive system, the midgut of freshwater shrimp Neocaridina davidi is composed of a tube-shaped intestine and the hepatopancreas formed by numerous caeca. Two types of cells have been distinguished in the intestine, the digestive cells (D-cells) and regenerative cells (R-cells). The hepatopancreatic tubules have three distinct zones distinguished along the length of each tubule—the distal zone with R-cells, the medial zone with differentiating cells, and the proximal zone with F-cells (fibrillar cells) and B-cells (storage cells). Fasting causes activation of cell death, a reduction in the amount of reserve material, and changes in the mitochondrial membrane potential. However, here we present how the concentration of ROS changes according to different periods of fasting and whether re-feeding causes their decrease. In addition, the activation/deactivation of mitochondrial superoxide dismutase (MnSOD) was analyzed. The freshwater shrimps Neocaridina davidi (Crustacea, Malacostraca, Decapoda) were divided into experimental groups: animals starved for 14 days, animals re-fed for 4, 7, and 14 days. The material was examined using the confocal microscope and the flow cytometry. Our studies have shown that long-term starvation increases the concentration of free radicals and MnSOD concentration in the intestine and hepatopancreas, while return to feeding causes their decrease in both organs examined. Therefore, we concluded that a distinct relationship between MnSOD concentration, ROS activation, cell death activation and changes in the mitochondrial membrane potential occurred.

Introduction

An animal organ which is exposed to various stressors originating from the environment is the midgut (the middle region of the digestive system). It takes part in maintenance of homeostasis of the entire organism. In nature, organisms can be exposed to periodic lack of food or various external harmful stressors, and as a result have evolved mechanisms that help animals to survive in adverse environmental conditions. The midgut is the main place of accumulation of reserve material in invertebrates, which may be used during starvation (Cervellione, McGurk & Van Den Broeck, 2017). Therefore, the processes related to the survival strategy during exposure to starvation are observed primarily in this organ. Long-term periods of starvation can cause numerous changes at the physiological, biochemical and molecular levels which lead to increased ability to survive (Wilczek et al., 2014; Lipovšek et al., 2015; Lipovšek et al., 2018; Lipovšek & Novak, 2015; Włodarczyk et al., 2017; Włodarczyk, Student & Rost-Roszkowska, 2019). Starvation can affect and damage many organelles, including the mitochondria (Ratcliffe & King, 1969; Włodarczyk et al., 2017), which are responsible for e.g., synthesis of ATP and reactive oxygen species (ROS), or activation of cell death (Martin, 2010; Kamogashira, Fujimoto & Yamaosba (2015); Włodarczyk, Student & Rost-Roszkowska, 2019). Ultrastructural alterations of mitochondria (Fernández-Checa, 2003; Faron et al., 2015; Włodarczyk et al., 2017) also cause changes in the functioning of the enzyme system. In several places along the mitochondrial respiratory chain (mainly due to complexes I and III), electrons can react directly with oxygen or another electron acceptor and generate free radicals. As a result, the superoxide anion radical (O2−), hydroxide ions (OH−), and hydrogen peroxide (H2O2) are formed. They must be eliminated by activation of the precise enzymatic system (Cadenas & Davies, 2000; Ramalho-Santos et al., 2009; Yao et al., 2004; Yao et al., 2007). Among the antioxidant enzymes, the superoxide dismutases (SOD) play protective roles against the effect of free radicals on organelles, e.g., mitochondria (Zelko, Mariani & Folz, 2002; Zhang et al., 2007; Combelles & Gupta, 2009; Umasuthan et al., 2012; Faron et al., 2015). Based on associated metal cofactors, four classes of SOD have been distinguished: Cu/ZnSOD, MnSOD, FeSOD and NiSOD (Fridovich, 1995; Zelko, Mariani & Folz, 2002). However, the Cu/ZnSOD commonly present in vertebrates has been replaced by MnSOD in crustaceans that is connected with the transport of oxygen by copper from haemocyanin (Brouwer et al., 2003). Two types of this enzyme can be recognized: cytMnSOD and mtMnSOD. mtMnSOD has been described as commonly distributed in animals, plants and bacteria, while cytMnSOD has been found only in Crustacea such as prawns, crabs, lobsters, and shrimps (Brouwer et al., 2003; Lin et al., 2010; Gómez-Anduro et al., 2012).

The digestive system of malacostracan Crustacea is composed of the tube-shaped ectodermal fore- and hindgut, while the endodermal midgut can be differentiated into intestine and hepatopancreas (Herrera-Álvarez, Fernández & Benito, 2000; Sousa & Petriella, 2006; Sonakowska et al., 2015; Sacristán, Nolasco-Soria & López Greco, 2014; Sacristán et al., 2016; Cervellione, McGurk & Van Den Broeck, 2017). Many studies have been conducted on crustaceans that have been starved, but they mainly concern the physiological alterations in organisms and they were conducted on species that are adapted to short-term periods of starvation, connected with e.g., molting or to long-term starvation (Sacristán et al., 2016). However, our previous studies on freshwater shrimp Neocaridina davidi (formerly Neocaridina heteropoda) were focused on long-term starvation. This species originates from Taiwan and it gained its popularity thanks to ease of breeding. In addition, its natural environment and feeding habitats resemble those observed in the majority of freshwater crustaceans all over the world. The midgut of N. davidi is composed of a tube-shaped intestine and the hepatopancreas formed by numerous caeca (Sonakowska et al., 2015). Two types of cells have been distinguished in the intestine, the digestive cells (D-cells) and regenerative cells (R-cells), while the structure of the hepatopancreas is more complicated. It is formed by numerous tubules with three distinct zones distinguished along the length of each tubule—the distal zone with R-cells, the medial zone with differentiating cells, and the proximal zone with F-cells (fibrillar cells) and B-cells (storage cells) (Sonakowska et al., 2015; Sonakowska et al., 2016). The impact of fasting and re-feeding on ultrastructural changes and activation of cell death in the midgut epithelium of this species has also been described (Włodarczyk et al., 2017; Włodarczyk, Student & Rost-Roszkowska, 2019). Fasting causes activation of cell death, a reduction in the amount of reserve material, and changes in the mitochondrial membrane potential. These alterations are probably the mechanisms which enable an animal to survive. However, re-feeding reverses all these changes (Włodarczyk et al., 2017; Włodarczyk, Student & Rost-Roszkowska, 2019). In order to gain a full view of the described changes, we decided to investigate how the concentration of ROS changes according to different periods of fasting and, what is the most important, whether re-feeding causes their decrease. In addition, the activation/deactivation of one of the stress-responsive factor important in antioxidative processes—mitochondrial superoxide dismutase (MnSOD)—was analyzed.

Material & Methods

Materials

The research was conducted on adult specimens of the freshwater shrimp Neocaridina davidi (formerly named as N. heteropoda) (Crustacea, Malacostraca, Decapoda). The specimens were obtained from local shrimp breeders and kept in a laboratory breeding facility, i.e., a 40 L shrimp tank equipped with heater with thermostat and mechanical filtration system. The water temperature was set to 21 °C, pH to 7 and total water hardness was 10 °d. The N. davidi shrimps were fed with JBL Novo Prawn. For the experiment, adult shrimps with cephalothorax length over 2.5 mm were chosen. The specimens were in good condition, actively moving and taking in food. To collect the material, no specific permissions were required for locations/activities.

Experiment

The fasting experiment was performed by placing shrimps in isolated plastic (250 mL) containers. Every day 10% of the water amount was replaced and the plastic containers were cleaned of excrements. Containers were kept in a shaded room to avoid development of algae. Shrimps were starved for 14 days. Specimens were collected for studies. Additionally, some specimens from the experimental group were re-fed for 4, 7, and 14 days. The periods of starvation and re-feeding were established according to our previous experiments and the results obtained (Włodarczyk et al., 2017; Włodarczyk, Student & Rost-Roszkowska, 2019). The number of specimens from the experimental group that were collected for the experiment and all techniques used are presented in Table 1. Individuals of N. davidi were slightly anesthetized on ice and midguts were dissected.

Table 1 Number of adult specimens of N. davidi used in the each part of the experiment.

	Confocal microscopy	
	Flow cytometry-DHE	MnSOD	DHE	
Number of days of starvation	Number of specimens analyzed Part 1: starvation	
Control	24	4	4	
14 days	24	4	4	
Number of days of re-feeding after 14 days of starvation	Number of specimens analyzed Part 2: re-feeding after 14 days of starvation	
4 days	24	4	4	
7 days	24	4	4	
14 days	24	4	4	

Methods

Confocal microscopy

Dihydroethidium (DHE).

A dye commonly used to evaluate reactive oxygen species (ROS) production, which penetrates all cell membranes. Isolated organs, without fixation, were washed in PBS (phosphate-buffered saline) with 0.0025% Triton X100 (RT) and stained with 30 µM DHE (Invitrogen) prepared from the 30 mM stock solution of DHE in DMSO. Tissues were incubated with the dye for 15 min in a dark chamber, at room temperature. After washing the material with PBS, it was labeled with DAPI (30 min in darkness). The material was analyzed with an Olympus FluoView FV1000 confocal microscope.

Superoxide dismutase (SOD) detection.

One of the primary antioxidant enzymes: increased MnSOD protects normal tissue against oxidative stress. MnSOD as one of the SOD enzymes is a critical antioxidant enzyme residing in mitochondria. The isolated organs (intestine and hepatopancreas) were fixed in Karnovsky fixative (2 h, 4 °C) and then permeabilized in PBS/0.1% v/v Triton X-100 pH 7.4 for 5 min at room temperature. In addition, tissues were blocked in PBS/5% w/v BSA pH 7.4 for 20 min and stained with primary antibody: anti-MnSOD rabbit polyclonal antibody (1:500; Stressgen) overnight at room temperature. Tissues were washed with PBS (pH 7.4) and incubated with goat anti-rabbit IgG secondary antibody conjugated with Alexa Fluor 488 (1:1000, Invitrogen). After washing the material with PBS, it was labeled with DAPI (30 min in darkness). The slides were analyzed with an Olympus FluoView FV1000 confocal microscope.

Sample preparation for Western blot analysis

Individuals of N. davidi from the control group were slightly anesthetized on ice and midguts were dissected (5 per sample). The midguts were then homogenized on ice in TBS buffer (Tris-buffered saline). Homogenates were then centrifuged at 4 °C, 15,000 g for 10 min. In the supernatants, total protein concentration was measured (Bradford, 1976) and detection of superoxide dismutase (MnSOD) was performed.

Western blot analysis

Denatured samples (water bath, 5 min, 95 °C) of identical amounts of protein (25 µg) were loaded and separated by 10% SDS-PAGE (30 min at 90 V, then 1 h at 120 V) and then transferred to the nitrocellulose membrane (Optitran BA-S 85, Whatman) with Mini Transfer-Blot (BIO-RAD) (2 h at 150 V, 300 mA). Next, the membranes were blocked (3% bovine serum albumin (BSA) in Tris-buffered saline (TBS), 1 h, at room temperature (RT)). Blots were incubated with specific primary antibody: anti-superoxide dismutase (MnSOD) developed in rabbit (Sigma) (overnight, at 4  ∘C, with continuous shaking). After incubation, the membranes were washed four times for 5 min in TBS with 0.1% Tween-20 (TBST) and then incubated with secondary antibody: Goat anti-rabbit IgG, AP conjugate (Enzo Life Sciences) (1 h, at RT, continuously shaking). Dilutions of the antibodies were conducted following the manufacturer’s instructions, in 1% BSA in TBS. After washing (4 × 5 min in TBST), the antibody complex was visualized by BCIP/NBT Solution (BioShop), washed again in distilled water, dried, and scanned.

Total protein concentration

Total protein concentration was measured according to the Bradford method (1976). The method is based on the binding of aromatic amino acids to the Coomassie Brilliant Blue (CBB, G-250, Sigma) dye with the v/v 1 (sample): 50 (CBB solution) ratio. The absorbance was measured at the wavelength of 595 nm, and the color intensity is proportional to protein concentration. The protein concentration was calculated from the calibration curve prepared from the absorbance measurements of the bovine serum albumin (protein content > 95%, Sigma) solutions of known concentrations (Bradford, 1976).

Flow cytometry

The dissected organs isolated from specimens from each experimental group were mechanically fragmented with scissors and suspended in 100 µL of PBS (pH 7.4). Then, the intestine and hepatopancreas cells were separated by gentle shaking in a homogenizer (Minilys; Bertin Technologies). The cell suspension was washed using centrifugation at 1500 rpm for five minutes and the precipitate was suspended in 100 µL of PBS buffer.

For the quantitative measurements of cellular populations undergoing oxidative stress were used the Muse Oxidative Stress Kit (Merck Millipore, No. MCH100111). The assay is based on dihydroethidium (DHE), which upon reaction with superoxide anions undergoes oxidation, resulting in red fluorescence. According to the manufacturer’s protocol, the results were expressed as the percentage of two populations of cells: ROS negative (live cells) and ROS positive (cells exhibiting ROS). The measurements were performed using the Beckman Coulter Instrument FC 500 flow cytometer with a 488 nm argon laser.

Statistical analysis

Statistical analyses were performed using the STATISTICA 10.0 software package (version 10.0; StatSoft, Tulsa, OK, USA; http://www.statsoft.com). Normality was checked using the Shapiro–Wilk test. The data were tested for homogeneity of variance using Levene’s test of equality of error variances. The significance of the differences in the percentage of ROS positivity between organs within the complementary groups was assessed using Student’s t-test, p < 0.05. The significance of differences in the percentage of ROS positivity among different time periods of starvation and re-feeding after starvation within each organ was assessed using the Tukey test, p < 0.05. All assays were based on 5–6 samples, performed in duplicate.

Results

Our previous studies have shown that there are no differences in the structure and changes in the intestinal epithelium of females and males (Włodarczyk et al., 2017; Włodarczyk et al., 2017). Therefore, these studies represent the results with the omission of N. davidi sexes. The use of dihydroethidine (DHE) for N. davidi intestine and hepatopancreas revealed a diverse distribution of ROS in all experimental groups. A weak signal was seen in some of the cells in both organs in the control group (Figs. 1A–1B). The quantitative analysis showed 2.8% ± 1.2 and 1.3% ± 0.6 ROS-positive cells in the hepatopancreas and intestine respectively (Table 2). After 14 days of starvation the percentage of ROS-positive cells strongly increased: 13.2% ± 2.1 in the hepatopancreas and 12.7% ± 1.2 in the intestine (Table 2). The qualitative analysis confirmed this, showing strong signals in both organs examined (Figs. 1C–1D). Re-feeding for 4 days after 14 days of starvation caused an increase in the number of ROS-positive cells in the hepatopancreas 15.7% ± 4.4, while in the intestine their number decreased to (9.2% ± 4.3) in comparison to animals starved for 14 days (Table 2). The signals from hepatopancreatic cells were stronger, whereas signals from intestinal cells were weaker according to the previous experimental group (Figs. 1E–1F). However, 7 days of re-feeding after starvation caused a strong decrease in the number of ROS-positive cells in both organs analyzed—5.4% ± 1.8 (hepatopancreas) and 2.0% ± 0.4 (intestine) (Table 2) –which was confirmed by the weak signals from epithelial cells in both organs (Figs. 1G–1H). Epithelial cells in hepatopancreas and intestine isolated from animals starved for 14 days and re-fed for 14 days also emitted weak signals (Figs. 2A–2B). The quantitative analysis showed that the number of ROS-positive cells in the hepatopancreas decreased in comparison to animals re-fed for 7 days (4.2% ± 0.6), while it was the same in the intestine: 2.0% ± 0.4 (Fig. 3) (Table 2).

Figure 1 3D representation of the DHE staining and DAPI staining of hepatopancreas and intestine.

ROS-positive cells (red), nuclei (n, blue). Confocal microscope. (A) A fragment of the hepatopancreas in non-starved animals. Scale bar = 20 µM. (B) A fragment of the intestine in non-starved animals. Scale bar = 20 µM. (C) A fragment of the hepatopancreas in animals starved for 14 days. Scale bar = 20 µM. (D) A fragment of the intestine in animals starved for 14 days. Scale bar = 30 µM. (E) Hepatopancreas in animals re-fed for 4 days after 14 days of starvation. Scale bar = 20 µM. (F) intestine in animals re-fed for 4 days after 14 days of starvation. Scale bar = 20 µM. (G) Hepatopancreas in animals re-fed for 7 days after 14 days of starvation. Scale bar = 20 µM. (H) Intestine in animals re-fed for 7 days after 14 days of starvation. Scale bar = 20 µM.

Table 2 Mean (x) ± standard deviation (SD) of cells with signs of DHE in the entire intestine and proximal zone of hepatopancreatic epithelium in N. davidi.

The different letters (a, b) denote significant differences between organs within the complementary groups (Student t-test, p < 0.05; n = 5).

	Hepatopancreas	Intestine	
Control group	2.8 ± 1.2a	1.3 ± 0.6a	
14 days of starvation	13.2 ± 2.1a	12.7 ± 1.2a	
4 days of re-feeding after 14 days of starvation	15.7 ± 4.4a	9.2 ± 4.3a	
7 days of re-feeding after 14 days of starvation	5.4 ± 1.8b	2.0 ± 0.4a	
14 days of re-feeding after 14 days of starvation	4.2 ± 0.6b	2.0 ± 0.4a	

Figure 2 3D representation of the DHE and DAPI staining of hepatopancreas and intestine.

ROS-positive cells (red), nuclei (n, blue). Confocal microscope. (A) Hepatopancreas in animals re-fed for 14 days after 14 days of starvation. Scale bar = 20 µM. (B) Intestine in animals re-fed for 14 days after 14 days of starvation. Scale bar = 20 µM.

Figure 3 Diagrammatic representation of the average percentage of ROS-positive cells in the hepatopnacreas and intestine during starvation and after re-feeding.

The different letters (a, b) denote significant differences between organs within the complementary groups (Student t-test, p < 0.05; n = 5). Cells were measured via flow cytometry.

The immunofluorescent method for detecting superoxide dismutase (MnSOD) at the level of the light microscope revealed a low level of this enzyme in the intestinal and hepatopancreatic cells in the control specimens of N. davidi. The specificity of the antibodies was confirmed by western blot technique (Fig. 4). The mitochondria of the epithelial cells in both organs in animals starved for 14 days expressed a higher amount of MnSOD in comparison to the control group. The longer the animals were re-fed after 14 days of starvation, the weaker were the signals emitted by epithelial cells in the hepatopancreas and intestine (Figs. 5A–5H, 6A–6B).

Figure 4 Western blot analysis of Superoxide Dismutase (MnSOD) in the midgut of freshwater shrimp Neocaridina davidi (25 µg of protein per each line).

Figure 5 3D representation of the MnSOD localization (green) and DAPI staining of hepatopancreas and intestine.

Nuclei (n, blue). Confocal microscope. (A) A fragment of the hepatopancreas in non-starved animals. Scale bar = 30 µM. (B) A fragment of the intestine in non-starved animals. Scale bar = 20 µM. (C) A fragment of the hepatopancreas in animals starved for 14 days. Scale bar = 30 µM. (D) A fragment of the intestine in animals starved for 14 days. Scale bar = 30 µM. (E) Hepatopancreas in animals re-fed for 4 days after 14 days of starvation. Scale bar = 30 µM. (F) Intestine in animals re-fed for 4 days after 14 days of starvation. Scale bar = 20 µM. (G) Hepatopancreas in animals re-fed for 7 days after 14 days of starvation. Scale bar = 30 µM. (H) Intestine in animals re-fed for 7 days after 14 days of starvation. Scale bar = 30 µM.

Figure 6 3D representation of the MnSOD localization (green) and DAPI staining of hepatopancreas and intestine.

Nuclei (n, blue). Confocal microscope. (A) Hepatopancreas in animals re-fed for 14 days after 14 days of starvation. Scale bar = 30 µM. (B) Intestine in animals re-fed for 14 days after 14 days of starvation. Scale bar = 20 µM.

Discussion

In recent years, intensive studies connected with the response of organisms to the stress of starvation/fasting in invertebrates have been carried out. In the studied invertebrate species, the authors described the susceptibility to starving and changes at the ultrastructural level in the epithelium of the digestive system (Wilczek et al., 2014; Lipovšek et al., 2015; Lipovšek et al., 2018; Lipovšek & Novak, 2015; Rost-Roszkowska, Janelt & Poprawa, 2018), including crustaceans (Cervellione, McGurk & Van Den Broeck, 2017; Pantaleão et al., 2015; Sacristán, Nolasco-Soria & López Greco, 2014; Sacristán et al., 2016; Włodarczyk et al., 2017; Włodarczyk, Student & Rost-Roszkowska, 2019). Ultrastructural changes may be associated with an increase in the concentration of free radicals in the examined cells (Kaminskyy & Zhivotovsky, 2014; Chen, Azad & Gibson, 2009; Redza-Dutordoir & Averill-Bates, 2016). Free radicals could derive either from numerous essential enzymatic and nonenzymatic reactions or can be caused by external stressors such as xenobiotics, X-rays, pathogens or even periods of starvation. Hence, the animals developed numerous defense mechanisms which participate in homeostasis maintenance. One of them is the production of antioxidants such as superoxide dismutases, catalase, glutathione, thioredoxin, etc. (Borković et al., 2008; Mailloux, 2018). When the balance between free radical generation and antioxidant defenses is disturbed, oxidative stress occurs (Bagchi & Puri, 1998; Combelles & Gupta, 2009; Ramalho-Santos et al., 2009; Lobo et al., 2010). Antioxidant non-enzymatic and enzymatic mechanisms are involved in the response to stressful conditions in crustaceans. Mainly two enzymes, catalase and superoxide dismutase (SOD), are treated in these aquatic invertebrates as the major indicators of oxidative stress (Borković et al., 2008; Mohana et al., 2016; Soberanes-Yepiz et al., 2018). The level of lipids and proteins in the diet of animals has an effect on the course of antioxidative processes (Goda, 2008; Sacristán et al., 2016; Méndez-Martínez et al., 2018b; Soberanes-Yepiz et al., 2018). Starved crayfish showed alterations in level of lipids, glycogen, and glutathione, but fasting did not affect the level of catalase, protein oxidation or activity of some enzymes. Long-term starvation also causes a decrease in the number of molts in crustaceans, suggesting that they do not adapt to long periods of fasting (Sacristán et al., 2016). The effect of diet on the activation of defense mechanisms against oxidative stress has been presented for e.g., Macrobrachium americanum (Soberanes-Yepiz et al., 2018), M. rosembergii (Mohana et al., 2016), Penaeus monodon (Sivagnanavelmurugan et al., 2014) and Cherax quadricarinatus (Sacristán et al., 2016). The transport of oxygen by copper from haemocyanin in crustaceans caused that Cu/ZnSOD has been replaced by MnSOD (Brouwer et al., 2003). Additionally, in these aquatic arthropods two types of this enzyme have been described: cytMnSOD and mtMnSOD. While mtMnSOD is commonly distributed in crustaceans as in the other animals, cytMnSOD has been only found in many species of prawns, crabs, lobsters, shrimps (Brouwer et al., 2003; Lin et al., 2010; Gómez-Anduro et al., 2012; Zhao et al., 2014; Soberanes-Yepiz et al., 2018). Total MnSOD in crustaceans is treated not only as a defense response against fasting, but also as an important factor in the immune responses against pathogen infections (Zhang et al., 2007; Yu et al., 2011), metal exposure (Haque et al., 2018) and even water pollution and ozonization (Oropesa, Floro & Palma, 2017). The relationship between oxidative stress and total MnSOD activation as the effect of starvation has also been described in starved specimens of N. davidi. Under the influence of two-week fasting, an increase in the concentration of free radicals from 2.8% and 1.3% to 13.2% and 12.7% (for the hepatopancreas and the intestine, respectively) and an increase in antioxidant (MnSOD) production were observed. In this study, the change in total MnSOD concentration was investigated, which could be an introduction to further studies. To learn about the regulation of antioxidative protection, future research on the MnSOD genes is necessary. However, one of the important stages of our experiment was the observation of antioxidative processes due to the re-feeding of animals after the period of starvation, which can lead to the death of half of the population. It should be mentioned that the period of 14 days of starvation and 4, 7, and 14 days of regeneration after returning to feeding were selected in accordance with our previous studies in which the PNR50 for N. davidi was presented (Włodarczyk et al., 2017; Włodarczyk, Student & Rost-Roszkowska, 2019). Differences in the values between the hepatopancreas and intestine during fasting are not statistically significant, so it can be concluded that the concentration of free radicals in both organs forming the midgut changes similarly. The results of our research suggest the occurrence of oxidative stress in the first stage of starvation and the activation of anti-ROS defense. In the initial stage of starvation, a rapid increase in the amount of free radicals leads to oxidative stress, which activates the defense mechanism in the form of antioxidant production as has been suggested for other crustaceans (Brouwer et al., 2003; Lin et al., 2010; Gomez & Anduro et al., 2006; Gómez-Anduro et al., 2012; Zhao et al., 2014; Sacristán et al., 2016; Soberanes-Yepiz et al., 2018). After reaching a high level of antioxidants, there is a gradual decrease in the concentration of free radicals caused by re-feeding. Differences in the concentration of free radicals between the hepatopancreas and the intestine after returning to feeding are statistically significant, but they are very small, which indicates that both organs react similarly. Our previous study also describes the effect of starvation and re-feeding of N. davidi on changes in ultrastructure and mitochondrial membrane potentials in hepatopancreatic and intestinal epithelial cells. Mitochondria are organelles which participate not only in ATP production, but also in synthesis of ROS, antioxidative enzymes, cell death activation, etc. (Fernández-Checa, 2003; Faron et al., 2015; Malota, Student & Światek, 2019). Additionally, these organelles can contain up to twelve sources of O2•–/H2O2 (Mailloux, 2018). The first signal of changes appearing in the mitochondria is the alteration in the transmembrane mitochondrial potential (ΔCm) (Faron et al., 2015; Sonakowska et al., 2016). Ultrastructural alterations together with transmembrane potential (ΔCm) may be connected with the activation of cell death (Sonakowska et al., 2016). We reported that starvation activates the degeneration of epithelial cells in N. davidi at the ultrastructural level and it causes an increase of cells with depolarized (non-active) mitochondria, while after re-feeding the mitochondria were regenerated at the ultrastructural level and the number of cells with active (polarized) mitochondria increased (Włodarczyk et al., 2017). Comparing the results of ROS activation and mitochondria degeneration, we can state that the increase in free radicals occurs together with a decreasing number of active mitochondria. The number of mitochondria with altered membrane potential also reaches a maximum after a period of 2 weeks of fasting (Włodarczyk et al., 2017). After re-feeding the shrimps, a decrease in the level of free radicals was observed as well as an increase in mitochondrial activity in the hepatopancreas and intestine. This may indicate an increase in electron leakage while reducing the mitochondrial membrane potential. Thus, increasing the production of free radicals does not have to be associated with greater mitochondrial activity (Speakman et al., 2004; Faron et al., 2015).

Depending on the level of ROS in the cell, different processes may proceed. At a low ROS level, the cell remains in a quiescent state, not dividing, and not differentiating. The increase in the level of ROS causes the beginning of proliferation, differentiation or even cell death. Therefore, the level of ROS in cells determines the maintenance of tissue homeostasis and repair of damaged tissues (Zhou, Shao & Spitz, 2014). Research in recent years (Karpeta-Kaczmarek et al., 2016; Dziewiecka et al., 2017) has shown the relationship between free radicals and cell death. Free radicals are an important element of signaling pathways of cell death processes. Oxidation of various chemical compounds by ROS leads to the release of e.g., cytochrome c from mitochondria, which is a signal that triggers apoptosis (Lobo et al., 2010; Kaminskyy & Zhivotovsky, 2014). Excessive concentration of ROS, in turn, causes oxidation of lipids, impairing the functioning of mitochondria, and decreases in ATP concentration, consequently causing necrosis. Cell death can also be activated by the first product of deactivation of superoxide ions, i.e., hydrogen peroxide (H2O2). If the enzymatic protection of the cell against H2O2 does not work, the Fenton reaction leads to the formation of toxic hydroxyl radicals (OH), against which the cell cannot defend itself. Hydroxyl radicals oxidize lipids in the membranes of various organelles, causing DNA damage and ultimately leading to apoptosis or necrosis (Chen, Azad & Gibson, 2009; Redza-Dutordoir & Averill-Bates, 2016). The relationship between the concentration of free radicals and cell death has been described in many organs of crustaceans (Menze et al., 2010; Wang et al., 2013) and it has also been presented due to an experiment that was carried out aimed at studies of the intensity of apoptosis during fasting and re-feeding (Włodarczyk, Student & Rost-Roszkowska, 2019). After two weeks of fasting, the intensity of apoptosis in the hepatopancreas and the intestine increases almost twofold, while after returning to feeding, regeneration takes place, so the intensity of apoptosis decreases. Two weeks after re-feeding, the intensity of apoptosis is close to zero. The excess of free radicals produced induces apoptosis, the maximum of which is for two weeks of starvation. During this time, the highest level of free radicals is also observed (13.2, 12.7% for the hepatopancreas and the intestine, respectively). However, the decrease in the intensity of apoptosis to the level of 0 after two weeks from the return to feeding is particularly interesting (Włodarczyk, Student & Rost-Roszkowska, 2019). This correlates with a decrease in ROS from 13.2 and 12.7% to 4.2 and 2% for the hepatopancreas and intestine, respectively, two weeks after re-feeding. Suspension of apoptosis after the regeneration period can be explained by the excessive level of antioxidants, which show a delay in relation to changes in free radical concentrations. The correlation between a high concentration of free radicals and the intensity of apoptosis is greater in the case of the intestine. In the case of the hepatopancreas, a significant increase in the amount of free radicals induces apoptosis to a lesser extent. It can be assumed that the hepatopancreas is the organ that has developed better defenses against free radicals (Borković et al., 2008; Goda, 2008; Méndez-Martínez et al., 2018b; Soberanes-Yepiz et al., 2018). However, in crustaceans total SOD activities were lower in this organ in comparison to gills and muscle (Borković et al., 2008).

Oxidative stress, and therefore the imbalance between ROS and antioxidants, has serious consequences for organisms. Free radicals as highly reactive compounds can cause DNA mutations and damage to genes responsible for the production of antioxidant proteins (Bagchi & Puri, 1998; Hensley et al., 2000; Golden, Hinerfeld & Melov, 2002; Faron et al., 2015). As a result, cells that are subjected to long-term oxidative stress can lose their defense against free radicals over time, by impairing the production of antioxidants. Superoxide dismutases (SODs) are enzymes which are responsible for the breakdown of the superoxide anion into oxygen and hydrogen peroxide.

It has been shown that starving shrimps causes a significant increase in the level of free radicals and a subsequent defense response in the form of an increase in the amount of antioxidants —here MnSOD. This means that the cells are subjected to strong oxidative stress, especially through the initial fasting period. The fasting can thus affect the impairment of the defense system against free radicals, and thus have adverse long-term effects. Regeneration after feeding starved shrimp can therefore be apparent because it does not take into account the irreversible changes that could have occurred in the cell’s DNA.

Conclusions

Studies on N. davidi shrimp have shown that: (a) long-term starvation increases the concentration of free radicals and MnSOD concentration in the intestine and hepatopancreas; (b) return to feeding causes a decrease in free radicals and in the concentration of MnSOD in the intestine and hepatopancreatic; (c) a distinct relationship between MnSOD concentration, ROS activation, cell death activation and changes in the mitochondrial membrane potential can be observed.

Supplemental Information

Supplemental Information 1 Calculations of ROS positive and ROS negative cells in intestine and hepatopancreas in all experimental groups

Click here for additional data file.

Additional Information and Declarations

Competing Interests

Author Contributions

Data Availability

The authors declare there are no competing interests.

Agnieszka Włodarczyk conceived and designed the experiments, performed the experiments, analyzed the data, prepared figures and/or tables.

Grażyna Wilczek performed the experiments, analyzed the data, authored or reviewed drafts of the paper, approved the final draft.

Piotr Wilczek and Sebastian Student contributed reagents/materials/analysis tools.

Anna Ostróżka contributed reagents/materials/analysis tools, prepared figures and/or tables.

Monika Tarnawska analyzed the data.

Magdalena Rost-Roszkowska conceived and designed the experiments, analyzed the data, authored or reviewed drafts of the paper, approved the final draft.

The following information was supplied regarding data availability:

Raw measurements are available as a Supplemental File. The raw measurements were collected using flow cytometry and were used for quantitative analysis at the University of Silesia in Katowice.

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
