# Peer review of "Relationship between ROS production, MnSOD activation and periods of fasting and re-feeding in freshwater shrimp Neocaridina davidi (Crustacea, Malacostraca)"

_PeerJ, doi:10.7717/peerj.7399_

## Round 0.1 · original submission · Major Revisions

Dear Magdalena,

Thank you for your submission to PeerJ.

It is my opinion as the Academic Editor for your article - Relationship between ROS production, MnSOD activation and periods of fasting and re-feeding in freshwater shrimp Neocaridina davidi (Crustacea, Malacostraca) - that it requires a number of Major Revisions.

The English requires major editing

You should improve the discussion of oxidative stress, indicating examples of studies in crustaceans, to better discuss your findings in comparison to the literature.

Indicate that organisms may adapt to ROS levels, and perhaps include future studies which would be interesting, such as analysing other biomarkers for oxidative stress.

Further discuss SOD in crustaceans, many reviews are available
The sentence: "It can be assumed that the hepatopancreas is the organ that has developed better defenses against free radicals." should be supported by a reference, as this fact is well known.

Reviewer 1 ·

Basic reporting

1. Line 53, I suggest inserting the preposition "in" before the word adverse.

2. Line 55, I suggest inserting the article “the” before the word processes.

3. Line 62, It is suggested to improve the description of the function and response of mitochondria in the oxidative stress.

4. Line 68, It is suggest mentioning some components of the superoxide dismutase family and their importance in oxidative stress.

Experimental design

Line 163. It is suggested that the authors describe the number of fields analyzed per experiment.

Validity of the findings

There is no description of figure 3 in the Results section.

Additional comments

The experiments are well performed and the findings are are in agreement with previous studies. I suggest discuss the importance of N. davidi in other types of studies and as a model to understand oxidative stress at the cellular level due to long-term starvation.

Reviewer 2 ·

Basic reporting

The study is interesting. The authors analyzed the production of ROS and the presence of the MnSOD in freshwater shrimp after starvation and refeeding in hepatopancreas and intestine. They report that long term starvation provokes an increase in ROS and in MnSOD and after refeeding, the values decrease. The association of starvation with oxidative stress is a very interesting topic.
In general the ms is clear and well presented in all the sections.

Experimental design

The experimental design is appropriate and clearly presented. In materials and methods, it is important to inform that the antibody used for the inmunolocalization of MnSOD does not cross react with other proteins and it was validated in Western blots. Additionally, several crustaceans have a citosolic MnSOD in addition to the classical mitocondrial enzyme and the antibody used might be detecting the 2 MnSODs. . Check paper published by Dr. Brown group and also data for l. vannamei and chinese crab, here are the DOIs
DOI 10.1042/BJ20030272, DOI: 10.1016/j.cbpb.2012.03.003, DOI:10.4238/2014.November.11.8

Validity of the findings

The resultasaare well presented and clear. I am not sure that the format used for the tables is the one required by Peer J., since the title should be at the top and not at the bottom and this should be concise, while the additonal information presented shoul be a footnote .
In figure 2, in the legend, a sentence is duplicated.

Additional comments

The resultss are well presented and clear. I am not sure that the format used for the tables is the one required by Peer J., since the title should be at the top and not at the bottom and this should be concise, while the additonal information presented shoul be a footnote .
In figure 2, in the legend, a sentence is duplicated.
General comments for the authors
It is an interesting study that adds information about responses of aquatic animals to oxidative stress derived from starvation. Check spelling and gramar in the whole paper, for instance the inset in figure 3, “ intestine” is not spelled correctly

Reviewer 3 ·

Basic reporting

1. English: correct, not professional enough
2. Literature references: week
3. Structure of the article: Abstract and Introduction have unnecessary details, but relevant information are omitted. Figure and table legends are not appropriately described
4. Results are not relevant to hypotesis and discussion is rather speculative

Experimental design

Nothing original.
Not well described with some unexplained issues.
Tecnical and ethical standards not given.

Validity of the findings

Not satisfied.

Additional comments

The manuscript entitled “Relationship between ROS production, MnSOD activation and periods of fasting and re-feeding in freshwater shrimp Neocaridina davidi (Crustacea, Malacostraca)" by Włodarczyk et al. represents the study of the digestive system in Crustacea where the authors use transmission electron microscopy and histochemical methods to detect and localize the ROS positive cells and MnSOD enzyme after periods of fasting and refeeding in shrimps.

There are to many speculative conclusions based on the scarce data.
The major flaw of this manuscript is that the authors insist on oxidative stress and elevated MnSOD activity after different periods of refeeding in shrimps.
Oxidative stress is much more than simple increase of ROS positive cells and localization of MnSOD.
Although mitochondria are the main intracellular source of ROS generation, these ROS also have physiological function, likeH2O2 which is used as a secondary messenger to coordinate oxidative metabolism with changes in cell physiology.
At least, authors could measure the activity of MnSOD, not just concentration, because elevated concentration doesn't seem its elevated activation. Some proteins can be synthesized in large quantities, but inactivated (some protein modifications)at some stage of their activation.
In addition, mitochondria are enriched with the antioxidant defenses required to degrade ROS, beside MnSOD, glutathione (GSH), thioredoxin (TRX) as well as CuZnSOD settled in its intermembrane space.
One should analyze a battery of antioxidants to be relevant for discussion on oxidative stress.
MnSOD and CuZnSOD catalyze the dismutation of O2•− to H2O2. Although mitochondria do produce O2•−, the dominant ROS in the matrix environment is H2O2 (Ryan J. Mailloux, “Mitochondrial Antioxidants and the Maintenance of Cellular Hydrogen Peroxide Levels,” Oxidative Medicine and Cellular Longevity, vol. 2018, Article ID 7857251, 10 pages, 2018.)

It is known that crustaceans experience starvation periods during their growing process as a result of sequential molting. Like authors emphasized, the starved condition has pro-oxidant effects due to the reduction of the antioxidant defense levels. But what about the adaption of these organisms?
Also it is not clear, why authors used both females and males in their study (how many females and how many males). Is there any sex difference in response to starvation?
Certainly it is not enough to argue the results obtained on just four or five individuals.

Overall, I don't recommend this article to be published in Peer J.

---

## Round 0.2 · accepted · Accept

Dear Dr. Rost-Roszkowska,

Thank you for your submission to PeerJ.

I am writing to inform you that your manuscript - Relationship between ROS production, MnSOD activation and periods of fasting and re-feeding in freshwater shrimp Neocaridina davidi (Crustacea, Malacostraca) - has been Accepted for publication. Congratulations!

Reviewer 1 ·

Basic reporting

In the introduction section the observations that I suggested were taken into account, so now it is clearer, the description of the oxidative stress in Neocaridina davidi is better understood.

The components of the superoxide dismutase family and their importance in response to oxidative stress were included as part of the introduction.

Experimental design

They describe in the section of material and methods the number of samples analyzed in this study.

Validity of the findings

The description of the results of figure number 3 are now described in the results section.

Additional comments

In general, the manuscript describes much better the experimental work done by the authors. In this article, they describe the importance of Neocaridina davidi as a biological model to understand the enzymatic role in oxidative stress under starvation conditions.